# Short-Term Exposure of *Dactylis glomerata* Pollen to Atmospheric Gaseous Pollutants Is Related to an Increase in IgE Binding in Patients with Grass Pollen Allergies

**DOI:** 10.3390/plants12010076

**Published:** 2022-12-23

**Authors:** María Fernández-González, Helena Ribeiro, Fco. Javier Rodríguez-Rajo, Ana Cruz, Ilda Abreu

**Affiliations:** 1Department of Plant Biology and Soil Sciences, Faculty of Sciences, University of Vigo, 32004 Ourense, Spain; 2Earth Sciences Institute (ICT), Pole of the Faculty of Sciences, University of Porto, 4169-007 Porto, Portugal; 3Department of Geosciences, Environment and Spatial Plannings, Faculty of Sciences, University of Porto, Rua do Campo Alegre 687, 4169-007 Porto, Portugal; 4Clinical Pathology Service, Immunology Laboratory Vila Nova de Gaia Hospitalar Centre, 4434-502 Vila Nova de Gaia, Portugal; 5Department of Biology, Faculty of Sciences University of Porto, 4169-007 Porto, Portugal

**Keywords:** *Dactylis glomerata* pollen, NO_2_, O_3_, atmospheric pollution, IgE reactivity, ELISA

## Abstract

The concentrations of nitrogen dioxide (NO_2_) and tropospheric ozone (O_3_) in urban and industrial site atmospheres are considered key factors associated with pollen-related respiratory allergies. This work studies the effects of NO_2_ and O_3_ on the protein expression profile and IgE binding in patients with grass allergies to *Dactylis glomerata* pollen extracts. Pollens were collected during the flowering season and were exposed to NO_2_ and O_3_ in a controlled environmental chamber. The amount of soluble protein was examined using the Bradford method, and the protein expression profile and antigenic properties were analysed using the immunoblotting and enzyme-linked immunosorbent assay (ELISA), respectively. Our results showed apparent inter-sera differences concerning the number and intensity of IgE reactivity, with the most prominent at bands of 55 kDa, 35, 33, and 13 kDa. In the 13 kDa band, both gases tend to induce an increase in IgE binding, the band at 33 kDa showed a tendency towards a reduction, particularly pollen exposed to O_3_. Reactive bands at 55 and 35 kDa presented an increase in the IgE binding pattern for all the patient sera samples exposed to NO_2_, but the samples exposed to O_3_ showed an increase in some sera and in others a decrease. Regarding the ELISA results, out of the 21 tested samples, only 9 showed a statistically significant increase in total IgE reactivity after pollen exposure to the pollutants. Our study revealed that although airborne pollen allergens might be affected by air pollution, the possible impacts on allergy symptoms might vary depending on the type of pollutant and the patient’s sensitisation profile.

## 1. Introduction

Airborne pollen is emitted into the air as part of the reproductive cycle of flowering plants. While airborne, it can be inhaled, and its allergens can trigger a hypersensitive reaction, leading to allergic symptoms in sensitive people, such as rhinitis, conjunctivitis, or, in the most severe cases, asthma attacks. The family Poaceae includes 12,000 species that are further classified into 771 genera belonging to 12 subfamilies [1]. This family’s importance lies in its high number of species, ecological function, and importance for the human population [2]. Being the most prolific and diverse family of plants growing in urban areas, its pollen can represent a significant seasonal component of the bioaerosol [3]. Owing to their ubiquity and high pollen production, they are considered one of the major causes of allergies, with the estimation that at least 40% of people with type I hypersensitive reactions are sensitised to grass pollen [4,5]. Several genera of the Pooideae subfamily, such as *Dactylis*, *Phleum*, and *Lolium*, are identified as the primary source of pollen aeroallergens. Allergens belonging to group 1 are the most known worldwide due to their abundance and potency and are also considered immunodominant among the grass pollen allergenic proteins [6].

In recent years, an increase in the cases of pollen-related asthma and allergy has been observed worldwide [7,8]. Moreover, several authors reported that the allergenicity to grass pollen was higher in urban areas than in rural areas [9,10,11,12]. This fact can be linked to the interaction of pollutants with the pollen and, thus, several studies have reported the modifications induced by air pollution, which have also been revised by Sénéchal et al. [13] and Ulrike and Dieter [14].

Air pollution’s chemical and biological components are often linked to anthropogenic activities [15]. In urban areas, particularly in densely populated cities, the most common pollutants in the atmosphere include airborne particulate matter, especially PM_10_ and PM_2.5_, nitrogen oxides (NOx), and tropospheric ozone (O_3_). The primary sources of NOx (NO + NO_2_) are high-temperature combustion processes, such as automobile exhaust, biomass burning, and other industrial processes. Nitrogen dioxide (NO_2_) is a reactive gas mainly formed by the oxidation of nitric oxide (NO), except in the case of traffic emissions, where NO_2_ is released by diesel engine vehicles [16]. The tropospheric ozone (O_3_) is one of the most critical secondary atmospheric pollutants worldwide, which is formed through a series of photochemical reactions involving nitrogen oxides (NOx), carbon monoxide (CO), and many other volatile organic compounds acting as precursor gases [16].

The interaction between airborne pollen and air pollutants can lead to changes in the reproductive function, physicochemical characteristics, and allergenic potential of the pollens inducing higher health risks [13]. Several authors have previously studied the effects of gases, such as NO_2_ and O_3,_ on pollen grains either in field studies, in controlled growth experiments, or pollen exposure experiments; variations in the expression of proteins, lipids, carbohydrates, and pollen wall surface components, were observed in these studies [12,13]. It has also been reported that the presence of these pollutants at higher concentrations can increase the reactivity of specific human IgE towards the pollen extracted from several species, which in turn could trigger lifestyle-related allergic reactions [17,18]. Different studies pointed out that in the most industrialised countries, asthmatic-sensitised people are at increased risk of asthma attacks due to ozone, nitrogen dioxide, sulphur dioxide, and inhalable particulate matter exposure. The concomitant increase in pollutant concentrations and airborne allergens can induce a greater IgE-mediated response to aeroallergens and could explain the increased frequency of respiratory allergies and asthmatic crises [17]. Another observed effect is the increase in the amount of the allergenic protein, for example, for the *Lolium perenne* and *Secale cereale* species [19,20], and biochemical modifications in the cells (i.e., oxidative stress) were associated with consequent changes in the allergenic potential in sensitised patients [14,21,22,23,24].

Therefore, the high concentration of air pollutants can influence the allergen bioavailability and potency of the pollen grains and might further lead to an increase in the prevalence of asthma and chronic asthma [25]. Moreover, airborne grass pollens are usually present at higher levels during extended periods of the year, increasing allergen exposure [3,26,27,28]. Furthermore, since the severity of sensitisation depends on individual genetic predisposition, all individuals might not react in the same manner to these modifications induced by air pollutants in the pollen allergens. Since Porto is a densely populated city and the most common atmospheric pollutant gases present are nitrogen oxides (NOx) and tropospheric ozone (O_3_), we aimed to investigate the effect of short-term exposure of NO_2_ and O_3_ on the protein expression profile and the IgE binding in 21 different sera samples of patients allergic to *Dactylis glomerata* pollen extracts using immunoblotting and ELISA (Enzyme-Linked ImmunoSorbent Assay).

## 2. Results

The amount of *Dactylis glomerata* soluble protein increased significantly (*p* < 0.05) in the pollens that were exposed to NO_2_ (2100 ± 29 µg/mL) when compared to the blank sample (1922 ± 53 µg/mL) but not when the pollens were exposed to O_3_ (2061 ± 115 µg/mL) (Figure 1). However, the SDS-PAGE analysis of the soluble proteins extracted from the pollens demonstrated similar profiles between the blank and the NO_2_ and O_3_ exposed samples. After Coomassie staining, several bands ranging from 13 to 100 kDa were observed (Figure 2). However, bands within the 13–55 kDa range exhibited higher intensity for the pollen samples exposed to the pollutants.

The results of total IgE binding for the 21 sera samples derived from the patients allergic to the pollen using the soluble protein extracts obtained for indirect ELISA immunoassays and through 1D immunoblotting are presented in Figure 3. In general, we observed a tendency towards an increase in the total IgE reactivity towards the soluble protein extracts of the pollens that were exposed to the pollutant gases, which is demonstrated by the positive values obtained for the mean difference in the IgE binding between the O_3_ or NO_2_ exposed samples and the blank sample. In the Western blot analysis, only seven and three sera samples out of the 21 tested samples showed a decrease in the total IgE reactivity towards the pollen extracts after O_3_ and NO_2_ exposure, respectively, while in the ELISA results, five and eight sera samples showed the similar results, respectively (Figure 3).

However, the subsequent ELISA analysis allowed us to evaluate if the changes induced by the two pollutant gases in total IgE reactivity were statistically significant. Only 13 sera samples exhibited significant differences when compared with the blank sample. A cluster analysis applied to the ELISA results of these 13 sera samples showed a distinction of two sample groups: one that clustered the sera samples presenting a decrease in total IgE reactivity after pollutant exposure when compared with the blank sample (sera 3, 6, 8 and 20), and the other in which a significant increase after O_3_ or NO_2_ exposure was observed. However, among this last group, further subdivisions were detected: (i) the IgE reactivity in blank and O_3_ samples were similar and lower than the NO_2_ (sera 1, 2); (ii) the IgE reactivity in blank and NO_2_ samples were similar and lower than O_3_ (sera 12, 13, 26, 28); (iii) the IgE reactivity in the blank sample was lower than O_3_ and NO_2_ (sera 5, 7, 14) (Figure 4).

Regarding the individual patient sera samples in the 1D immunoblot analysis, the evaluation of the IgE reactive bands to the *Dactylis glomerata* pollen soluble protein extracts revealed several distinct bands, among which many bands were shared by different sera samples (Figure 5). However, the IgE binding pattern had inter-sera differences concerning the number and intensity of IgE reactive bands.

The most prominent bands showed a molecular weight of approximately 55, 35, 33, and 13 kDa. The percentages of sera samples reactive to these above-indicated groups were recorded as 57%, 100%, 95%, and 57%, respectively. Other bands were also detected in the range of 32 to 30 kDa, 25 to 20 kDa, and 70 kDa (Figure 5).

Focusing only on the prominent bands, we could observe that for the 13 kDa band, both gases tend to increase the IgE binding (mean differences higher than 0.528). On the contrary, the IgE reactive band observed at 33 kDa showed a tendency towards a reduction in the IgE binding to the pollen extracts of the polluted samples, particularly visible for the pollen sample exposed to O_3_ (negative mean difference of −0.959). However, the reactive bands at 55 and 35 kDa presented an increase in the IgE binding pattern for all the patient sera samples exposed to NO_2_, but for the samples exposed to O_3,_ the results balanced each other with either an increase or decrease, as the mean difference between the values for the IgE binding in O_3_ exposed and the blank samples was close to zero (Figure 6).

## 3. Discussion

Pollutants in the atmosphere of industrialised areas are increasing rapidly, which has become a stress factor for plants, affecting the physiological processes in their pollens [13,29]. The additional influence of urban pollution, such as O_3_ and NO_2,_ might prompt a more severe immunological response to the airborne pollen [30,31,32,33], either by distressing the respiratory mucosa and immune system or changing the potency of these pollen allergens [23,34].

The exposure of *Dactylis glomerata* pollen to NO_2_ or O_3_ can affect its proteomic features. Our results showed that the pollen samples exposed to NO_2_ had a significantly higher amount of soluble proteins when compared with the blank sample, but the samples exposed to O_3_ did not show such results. Differences in the amount of soluble protein after pollen exposure to pollutant gases were reported in several previous studies, but the outcome varies as the increase or decrease in the amount of protein was shown to depend on the plant species [35,36,37]. In this sense, a study carried out in Madrid city reported a positive correlation between Poaceae pollen loads and O_3_ concentration outside the clinically relevant season [38]; other research carried out in Bratislava pointed to a strong and significant positive correlation between Poaceae airborne pollen and several air pollutants (PM_10_, PM_2.5_, CO, O_3_, and NO_2_) [39], while the pollen allergenic potency was positively significantly correlated with NO_2_ levels [40]. In Mexico City, it was observed that the amount of *Fraxinus* and Cupressaceae pollen was correlated with increases in NO_2_, PM_10_, and PM_2.5–10_ [41]. NO_2_ was shown to react with the pollen grains, inducing the degradation of its structure, which further leads to protein amount changes [42]. On the other hand, O_3_ was reported to induce strong oxidative activity that affects the activity of biomolecules, such as proteins [43].

In the 1D immunoblots results, differences in IgE binding due to NO_2_ and O_3_ exposure were also observed. It has been reported that the effects of pollutants on pollen could be dependent on plant species and the type of allergen, not necessarily implying an association between the increase in the pollen allergen content and the allergenicity degree in the atopic patients [12,19,44,45,46,47,48]. In the present study, NO_2_ exposure led to an increase in the total and specific IgE binding bands (except one) of the patient sera towards the exposed pollen-soluble protein extracts. On the other hand, total IgE binding to soluble protein extracts from the pollen samples exposed to O_3_ increased, but the band-specific IgE binding was dependent on the reactive band. Oxidation reactions can lead to the degradation of allergenic proteins reducing their recognition by the immune system [23,24,49]. A dissimilar influence of O_3_ on pollen has been reported in experiments conducted in vitro and in vivo. For instance, in a study with pollens derived from birch trees exposed to environmental conditions, it was observed that elevated O_3_ concentrations revealed less immune-modulatory but higher immune-stimulatory potential [50]. In *Ambrosia* plants exposed to elevated O_3_ concentrations during the whole vegetation period, Kanter et al. [44] detected no such significant differences in the amount of Amb a1 but showed an increase in the transcript level of the allergens. The experiments performed by Pasqualini et al. [47] exposed ragweed pollen directly to elevated ozone levels for several days and showed that the Amb a1 allergen content did not significantly increase. However, Ribeiro et al. [37] reported that O_3_ increased IgE reactivity for Pla a1 and decreased for Pla a2 using the sera samples from patients allergic to *Platanus* pollen.

On the other hand, the effects of NO_2_ on the increase in the potency of pollen allergens have also been documented. Several in vitro studies carried out on pollen from *Phleum pratense* L. showed that NO_2_ concentrations between 500 ppb–5000 ppm could induce an increase in PCGs, reported as a dose-dependent increase in pollen grain damage [51], and a reduction of IgE binding to Phl p 2, 5b, and 6 [52]. In addition, Chassard et al. [42] showed a direct correlation between NO_2_-fumigated pollen with higher Th2 response in human cells, revealing changes in the protein content and pollen structure [42].

Also, in the 1D immunoblot analysis, the IgE binding pattern revealed inter-sera differences that could be related to sensitisation to each individual allergen, as most sera were derived from atopic patients sensitised to Poaceae mixture (gx1) or even associated with the IgE response cascade against grass pollen, which has been documented to evolve from the monomolecular to polymolecular level [53].

Poaceae pollen allergens are grouped by protein structures and functions and are further characterised by distinct allergen subsets, which show a strong cross-allergenicity between different species [54,55]. In the present study, we observed a reactive band at 33 kDa that exhibited reduced IgE binding to the protein extracts of pollen exposed to the pollutants. On the contrary, a 13 kDa band presented enhanced IgE binding due to exposure to pollutants. These results suggest that each pollutant influences the specific protein macromolecules and the post-translational protein modifications differently, which might point towards distinct molecular interactions between the allergens and the immune system. For example, stability effects influencing the accumulation and degradation of the allergenic proteins, generating new epitopes, modifying the existing epitopes, generating new adjuvant functions, or changing the existing adjuvant functions were reported previously [12,23]. In this case, for instance, the lipid-binding capacities could be related to modified ligand binding sites and the multiplication or shielding of epitopes or adjuvant functions occurring due to cross-linking of allergenic proteins [12,23].

In the present study, we observed that 55 and 35 kDa bands were associated with an increasing IgE binding trend due to the exposure to NO_2,_ but no changes in this trend were observed due to O_3_ exposure. In this last case, the number of tested sera demonstrating a reduced IgE binding trend was almost equal to the number of sera presenting an increase in this trend. This behaviour may be due to each reactive band being composed of more than one IgE pollen reactive peptide combined with the fact that most of the tested sera samples were from sensitised patients to Poaceae mixture (gx1), and therefore the presence/absence of reactivity to *D. glomerata* pollen allergen was unknown. Smiljanic et al. [12] compared timothy grass pollen allergens from plants grown in polluted and rural areas and observed IgE reactive bands composed of more than one allergen group, e.g., 35 kDa (Phl p 1 and Phl p 5) or 55 kDa (Phl p 4 and Phl p 13). Phl p4 was more abundant in pollen from polluted environments, while Phl p13 was less. Therefore, in each serum, the IgE reactive bands at 55 and 35 kDa might be the combination of only one or more IgE pollen reactive peptides that these gaseous pollutants might influence differently. Nowadays, the use of recombinant allergens in determining allergen-specific IgE levels is increasing, so the influence of different gaseous pollutants on IgE-binding from patient sera samples to specific pollen allergens can contribute to providing important information for designing precise medicine strategies.

Indirect ELISA results have allowed us to determine the statistical significance by monitoring the total IgE reactivity changes induced by each pollutant compared with the non-exposed sample. We observed a lower number of sera presenting a significant increase in the IgE reactivity after pollen exposure to the pollutants compared with the Western blot analysis results. Out of the 21 tested sera samples, only nine showed a statistically significant increase in IgE reactivity. This might be because the ELISA assay showed the total IgE reactivity of the sera samples to the protein extracts of *Dactylis glomerata* pollen, and, for instance, all the IgE reactive bands lower than the 14 kDa bands that were not present in the immunoblots could further contribute to the results.

Finally, the NO_2_ and O_3_ concentrations tested in our study corresponded to the limit value legislated by the European Union for human health protection according to the EU Directive 2008/50/EC. The revision of this directive, by adopting values closer to the ones proposed by the World Health Organization, could contribute to improving human health and positively impacting pollen allergenicity.

## 4. Materials and Methods

### 4.1. Pollen Samples

The *Dactylis glomerata* L. pollen was collected during the peak flowering period (June) in Porto city (north-western Portugal). The samples were collected on the same day and only from plants presenting dehiscent anthers. Afterward, the samples were dried at 24 °C in the laboratory, and then the pollens were recovered by passing through different grades of sieves to obtain pure pollens. Finally, the pollen samples were stored at −20 °C until further use.

### 4.2. Patient Sera

For ELISA and Immunoblot analyses, 21 sera from anonymous random atopic patients who were sensitised to *Dactylis glomerata* pollen extracts (g3) and/or Poaceae mixture (gx1—*Dactylis glomerata, Festuca elatior*, *Lolium perenne*, *Phleum pratense*, and *Poa pratensis*), determined ImmunoCAP^TM^ fluorescence enzyme immunoassay (FEIA) test (Thermo Fisher Scientific/Phadia AB, Uppsala, Sweden), were used. Briefly, the sera samples were separated from the whole blood and collected by venepuncture, and the allergen-specific IgE levels were measured using the ImmunoCAP™ (FEIA) test in a hospital in the Porto region. Information regarding the patients’ demographic data is unavailable due to the confidentiality agreement. However, the information regarding the class of specific IgE recognised and its values (kUA/L) is available. The 21 sera were selected to match the seven classes of specific IgE recognition (kUA/L): class 0 [<0.35]; class 1 [0.35–0.7]; class 2 [0.7–3.5]; class 3 [3.5–17.5]; class 4 [17.5–50]; class 5 [50,51,52,53,54,55,56,57,58,59,60,61,62,63,64,65,66,67,68,69,70,71,72,73,74,75,76,77,78,79,80,81,82,83,84,85,86,87,88,89,90,91,92,93,94,95,96,97,98,99,100]; Class 6 [>100], which indicated the allergenic severity to grass pollen allergens.

### 4.3. In Vitro Exposure to Gases Pollutants

*Dactylis glomerata* L. pollens were exposed to NO_2_ and O_3_ in a controlled environmental chamber equipped with a solar simulator (Newport Oriel 96000 150 W) and a fan (SUNON SF23080AF) to homogenise the air and temperature. Humidity sensors (EBRO EBI20) were also integrated inside this controlled environmental chamber to monitor meteorological conditions [56]. Pollen samples (150 mg of dry weight) were introduced inside a tube with both edges covered with 23 µm pore mesh (SEFAR PET 1000) that was positioned over a fan that impelled the air within the chamber to pass through the tube. Pollen samples were individually exposed to the gases for 6 h at the limit value recommended for the protection of human health in Europe (O_3_: 0.061 ppm and NO_2_: 0.025 ppm; European Union Directive 2008/50/EC of May 21 2008 on ambient air quality and cleaner air for Europe). To generate ozone, an A2ZS-1GLAB ozone system connected to a timer (OMRON H3DK-S1) to control the gas injection was used. The NO_2_ was generated in a sealed bottle through the chemical reaction carried out between concentrated nitric acid (to ensure the production of NO_2_). The blank pollen sample was subjected to the same exposure strategy; however, the air inside the chamber was not contaminated with the toxic gases.

### 4.4. Protein Extraction and Quantification

Pollen grains (50 mg) were suspended in 1 mL of phosphate-buffered saline (pH 7.4) at 4 °C. Soluble proteins from the pollen samples were extracted by continuous stirring for 4 h. The resulting suspension was centrifuged at 13,200 rpm for 30 min at 4 °C. The supernatant was then filtered using a 0.45 µm Millipore filter and centrifuged again at 13,200 rpm for 30 min at 4 °C. The quantification of the soluble proteins was conducted using the Bradford method [57]. A standard curve was calculated using different BSA concentrations (0–2000 μg/mL) and then used as a protein estimation standard. All the measurements were performed in triplicate.

### 4.5. ELISA Immunoassay

ELISA immunoassay was performed following the methods described by de Leon et al. [58] and Cases et al. [59] with slight modifications. Microtiter plates (Nunc-Immuno^TM^, 96 well plates) were coated with 15 µg/mL per well of the protein extracted from the pollens exposed to NO_2_ and O_3_. Then, the plates were incubated overnight at 4 °C in a moist chamber. The wells were blocked (200 µL/well) using PBS-BSA-T for 1 h at 37 ºC. Later the plates were incubated for 3 h at 37 °C with the 21 patient sera samples used in the immunoblot experiment. After six washes with 200 µL/well PBS-T, the plates were incubated with 100 µL/well of the anti-human antibody IgE-HRP (1:2000) for 1 h at 37 °C. Following this, the plates were washed three times with 200 µL/well of PBS-T and were incubated with 100 µL/well of o-phenylenediamine for 10 min at room temperature in the dark. The reaction was stopped using 50 µL/well of 3M H_2_SO_4_. Finally, the absorbance was measured at 492 nm wavelength, which revealed the total quantity of IgE left in the well. A blank sample containing only PBS was used as the negative control. In addition, a blank sample coated with patient sera was used as the positive control.

### 4.6. SDS-PAGE and Immunoblots

Proteins from pollen extracts (15 µg/mL per line) were separated using 12.5% (*w*/*v*) polyacrylamide gels under reducing conditions [60]. The proteins were visualised using Coomassie Brilliant Blue R-250 staining. The molecular weight (MW) of the protein bands of the samples was estimated by comparing them with the MW of an established protein marker (PageRuler Plus Prestained Protein Ladder).

Immunoblot analyses were performed following the protocol described by Sousa et al. [52]. Proteins from pollen extracts were separated in the 12.5% (*w*/*v*) polyacrylamide gels under the reducing condition (15 µg/mL per lane). Thereafter, the proteins were electrophoretically transferred to nitrocellulose membranes (Protran, Whatman^®^ Schleicher and Schuell, Germany) using the transfer buffer (192 mM glycine, 25 mM Tris, and 20% methanol) for 2 h at 200 mA. The membranes were saturated for 1 h with a blocking buffer [5% non-fat dry milk (*w*/*v*), 0.1% goat serum (*v*/*v*), and 0.1% Tween in 20 mM Tris, 150 mM NaCl (TBS)], and then were incubated for overnight at 4 °C with sera diluted at 1:10 in blocking buffer. After the washing step, blots were probed with the mouse anti-human IgE-HRP (1:10,000). The membranes were revealed using the Luminata Crescendo Western HRP substrate (Millipore) and visualised using the ChemiDocTM XR+ System. The antigenic profile bands of the immunoblots were quantified using the software, Image Lab 5.2 (Bio-Rad Laboratories).

### 4.7. Statistical Analysis

Having the blank sample as the reference, we also performed (i) the t-test to evaluate the quantitative effects of each pollutant gas separately in the pollen grains and (ii) the hierarchical cluster analysis using the indirect ELISA results to categorise the different tested sera based on the statistical significance of their quantitative differences. The number of clusters was also determined using (i) the squared Euclidean distance as a distance measure and (ii) the nearest neighbour method as a linking method.

Statistical analysis of all the experimental results was conducted using the Microsoft Office Excel 2013 spreadsheet and IBM SPSS statistics version 24. A *p*-value < 0.05 was considered as a measure of statistical significance.

## 5. Conclusions

Significant changes (increase) in the amount of the soluble protein in *D. glomerata* L. pollen were only induced by NO_2_.

Inter-sera differences regarding the number and intensity of IgE reactivity to pollen protein extracts were observed in samples either exposed or non-exposed to pollutant gases. Moreover, trends in IgE binding intensity (increase, decrease, or no change) after pollutants exposure were dependent on the pollutant and reactive band (allergens). Overall, a tendency showing that exposure to NO_2_ induced an increase in total and band-specific IgE binding. The influence of O_3_ was variable depending on each IgE reactive band. These divergent results might point towards distinct interactions between air pollutants, pollen allergens, and the immune system.

Our study revealed that while the airborne pollen allergens might be affected by air pollution, the impact on allergy exacerbation might vary depending on the type of pollutant and on the patient’s sensitisation profile.

## Figures and Tables

**Figure 1 plants-12-00076-f001:**
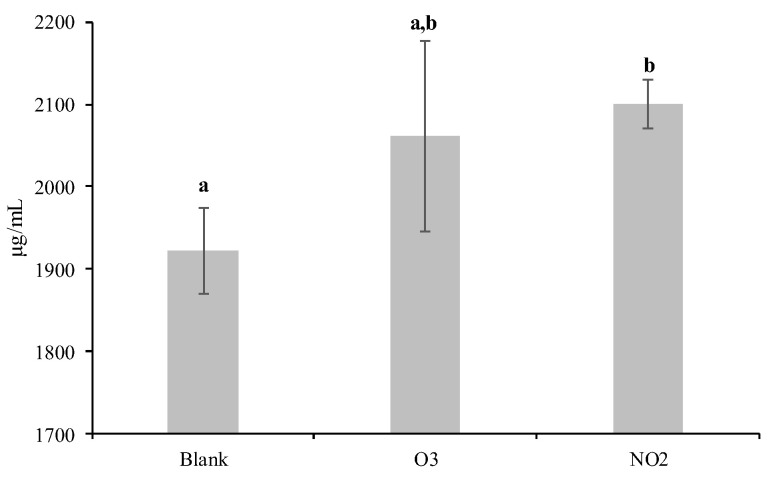
Average and standard deviation of the total soluble protein content of *Dactylis glomerata* pollen non-exposed to pollutants (Blank) and exposed to O_3_ and NO_2_ gaseous pollutants during 6 h at the limit value recommended for the protection of human health in Europe (O_3_: 0.061 ppm and NO_2_: 0.025 ppm; European Union Directive 2008/50/EC). Different letters indicate values statistically significantly different (ANOVA test followed by Tukey HSD post hoc, *p* < 0.05).

**Figure 2 plants-12-00076-f002:**
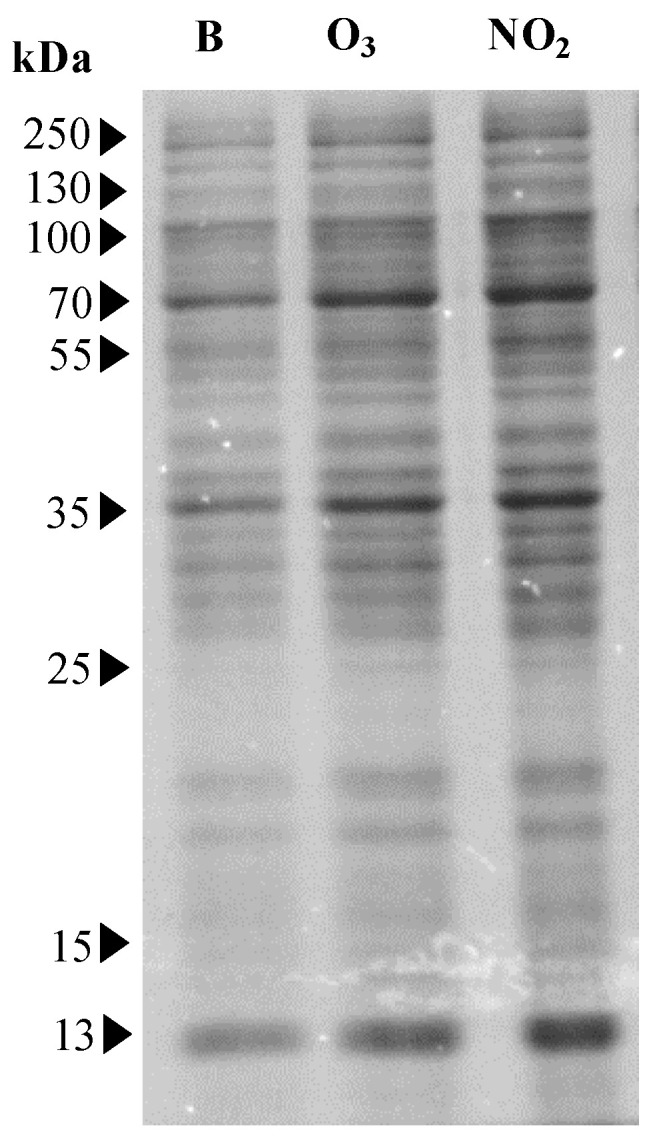
Results of SDS-PAGE gel of total soluble proteins from *Dactylis glomerata* pollen extracts non-exposed to pollutants (B; lane 1) and exposed to O_3_ (lane 2) and NO_2_ (lane 3) gaseous pollutants during 6 h at the limit value recommended for the protection of human health in Europe (O_3_: 0.061 ppm and NO_2_: 0.025 ppm; European Union Directive 2008/50/EC). The molecular weight of the bands is expressed in kDa.

**Figure 3 plants-12-00076-f003:**
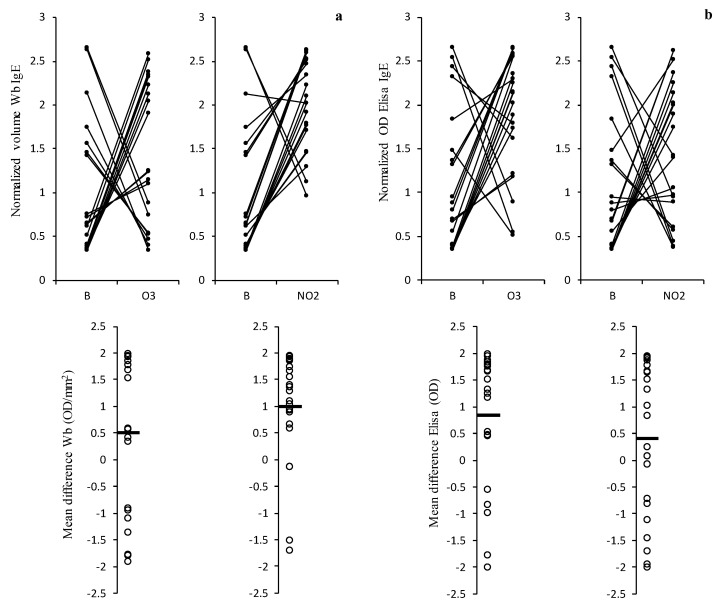
Results of the variation of total IgE binding normalised volume (OD/mm^2^) in the 1D Western blot (**a**) and normalised optical density (OD) in the Elisa essay (**b**) observed between non-exposed pollen (B) and exposed to pollutants (O_3_ or NO_2_) in 21 patient sera tested. Below each graph is presented the respective mean differences in IgE reactivity (expressed in OD/mm^2^ for Western blot, calculated based on each volume profile and their total area, and in OD for Elisa). The IgE values (OD/mm^2^ or OD) were normalised using the z-scores method based on the total optic density of the electrophoretic profiles in the Western blot. This value was obtained by dividing the volume between each band’s area for the Western blot and the volume between the optical densities for the ELISA method. Then the z-scores method was applied. The black horizontal bar in the graphs below indicates the average mean difference.

**Figure 4 plants-12-00076-f004:**
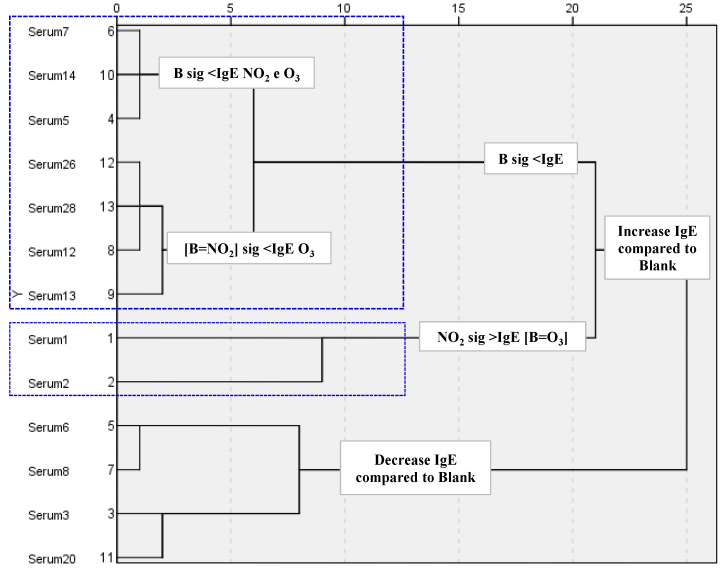
Hierarchical cluster analysis of the results from the indirect ELISA to categorise the different tested sera based on the statistical significance of their quantitative differences (number of clusters determined using the squared Euclidean distance as a distance measure and the nearest neighbour method as a linking method). Two groups of sera were formed based on an IgE increase compared to the blank sample (B) (1, 2, 5, 7, 14, 12, 13, 26, and 28 samples sera) and IgE decrease compared to the blank (3, 6, 8, and 20 samples sera), sig: significative.

**Figure 5 plants-12-00076-f005:**
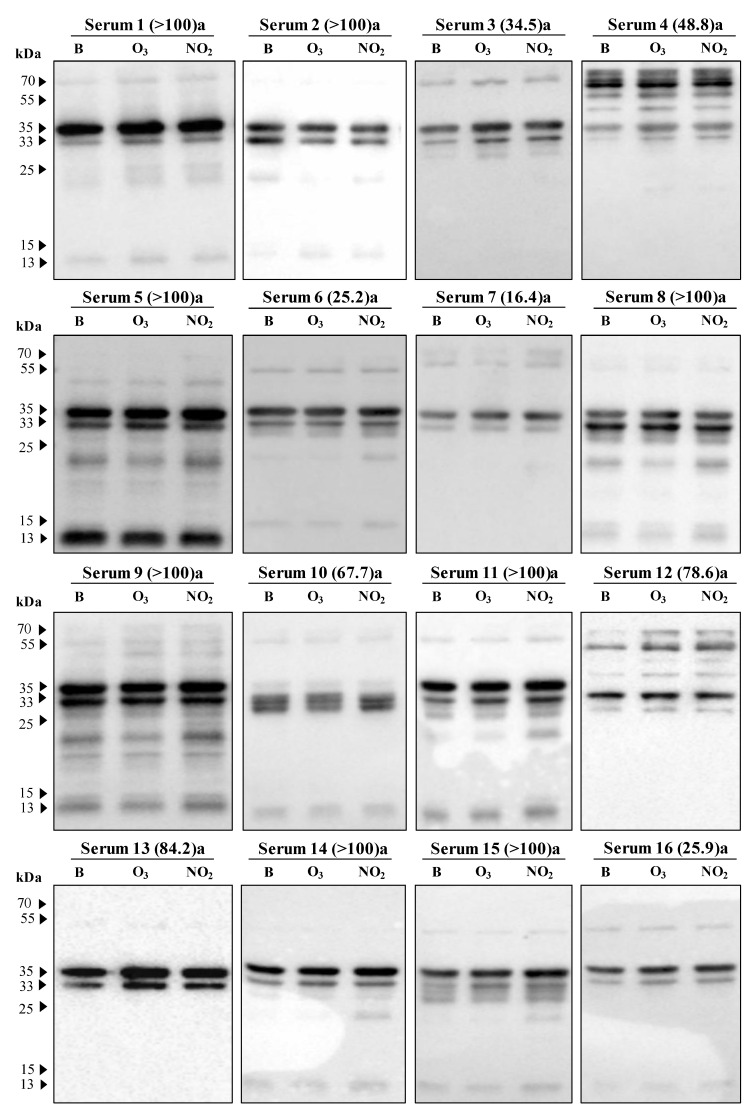
Immunoblot results from 21 sera of patients sensitised to grass pollen showing the IgE-reactive bands against protein extracts of *Dactylis glomerata* pollen non-exposed (B) and exposed to pollutants (O_3_ or NO_2_). At the top of each immunoblot in brackets is the serum IgE concentration (kUA/L), and the letter corresponding to the sensitisation (a: gx1— Poaceae mixture of *Dactylis glomerata, Festuca elatior*, *Lolium perenne*, *Phleum pratense*, and *Poa pratensis*; b: g3—*Dactylis glomerata*).

**Figure 6 plants-12-00076-f006:**
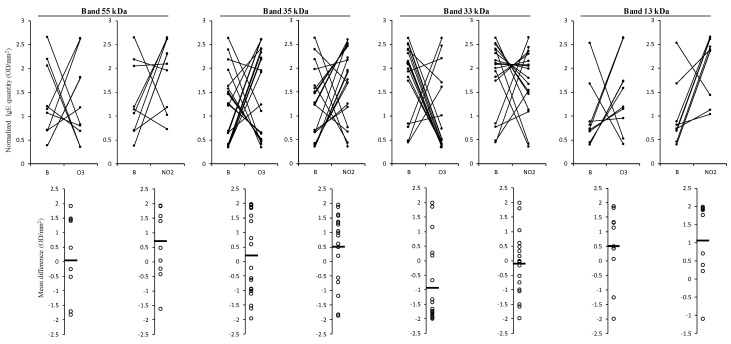
Results of the variation of total IgE binding normalised volume (OD/mm^2^) in the 1D Western blot observed between non-exposed pollen (B) and exposed to pollutants (O_3_ or NO_2_) for the four most common reactive bands (55, 35, 33, and 13 kDa) in the 21 patient sera tested. Below each graph is presented the respective mean differences in IgE reactivity (expressed in OD/mm^2^). The IgE responses of patients were normalised based on the content of electrophoretic profiles of non-exposed, O_3,_ and NO_2_ exposed pollen samples, using the z-scores method. The differences were expressed in OD/mm^2^, calculated based on each volume profile and total area.

## Data Availability

Not applicable.

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
