# Peer review of "Short-Term Exposure of Dactylis glomerata Pollen to Atmospheric Gaseous Pollutants Is Related to an Increase in IgE Binding in Patients with Grass Pollen Allergies"

_plants, 2022, doi:10.3390/plants12010076_

Round 1

Reviewer 1 Report

Interesting topic and research. Some corrections are suggested below.                    Line 44 : the statement is not correct , since the percentage does not refer to all type I allergies, but to pollen induced allergic rhinitis . Another corresponding reference has to be added.                                                               Line 72: please explain sporopollen ? The difference between hypersensitivity and allergy seems not clear .                                                                                 The Material and Methods section has to be placed after Introduction , followed by Results .

Author Response

The authors want to thank the work done with the manuscript.

Reviewer 2 Report

Ref.: plants-2041544-peer-review-v1

Title: Short-term exposure of Dactylis glomerata pollen to atmospheric gaseous pollutants is related to an increase in IgE binding in patients with grass pollen allergies

This Manuscript is focused on the Short-term exposure of Dactylis glomerata pollen to atmospheric gaseous pollutants is related to an increase in IgE binding in patients with grass pollen allergies. The manuscript is more or less well organized, contains relatively adequate relevant materials and language is good. However, after reading this manuscript, I found some minor problems. Some of the comments and suggestions are as follows:

1.      The introduction part should be more specific and clear about the airborne pollen and air pollutants. For example, the author wrote some specific information about the grass pollen but in the 1st paragraph, he can briefly describe an overall description about the airborne pollen. Not in details but just 2/3lines will be enough to convince a new reader about the airborne pollen.

In line 73-76 the author wrote, ‘It has also been reported that the presence of these pollutants at higher concentration can increase the reactivity of specific human IgE towards the pollen extracted from several species, which in turn could trigger lifestyle-related allergic reactions [17,18]’. Please specify some more points about the relationship between airborne pollen, IgE, and health risk, not only tell us to see the references.

Please write the elaboration or definition of ELISA (Line 91)

2.      Result section is well organized. However, some figures are not clear to see and understand at all (eg. Figure 2 and figure 5).

3.      In discussion section, the author should compare his result with some more published result. Moreover, the author should give some suggestions to way out of this situation.

4.      Materials and methods: Please specify the flowering season from which month to which month. Did the author collect the sample in one day or on different days?

What is Immuno CAPTM and Phadia AB? Please write the elaboration or definition of them. Someplace the author wrote ImmunoCAPTM and someplace ImmunoCAPTM (eg. Line 274 and 276). I suggest to read the manuscript thoroughly again and again and make sure that there is no wrong or fake information in the manuscript.

5.      In conclusion, please highlight your main invention, what actually your result shows. The main outcome is not very properly highlighted. I suggest you to rewrite the conclusion.

Finally, the paper can be accepted for publication after revising the above-mentioned comments.

Author Response

(The authors gave the same response as above.)

Round 2

Reviewer 1 Report

Line 42 : I suggest a more clear phrase : Owing to their ubiquity and high pollen production, the members of this family are considered as one of the major causes of seasonal allergies in the world, considering the estimated prevalence of pollen allergy  up to 40 % . Reference to be added : D'Amato G et al.  Allergenic pollen and pollen allergy in Europe. Allergy 2007, 62:976-990

Author Response

Q1-Line 42: I suggest a more clear phrase: Owing to their ubiquity and high pollen production, the members of this family are considered as one of the major causes of seasonal allergies in the world, considering the estimated prevalence of pollen allergy up to 40 %. Reference to be added: D'Amato G et al.  Allergenic pollen and pollen allergy in Europe. Allergy 2007, 62:976-990

R1- Following the reviewer suggestion we add the reference:

D'Amato, G.; Cecchi, L.; Bonini, S.; Nunes, C.; Annesi-Maesano, I.; Behrendt, H.; Liccardi, G.; Popov, T.; van Cauwenberge, P. Allergenic pollen and pollen allergy in Europe. Allergy 2007, 62, 976-990.
